# Discovery and construction of surface kagome electronic states induced by *p-d* electronic hybridization in Co₃Sn₂S₂

Li Huang [1,2,11], Xianghua Kong [3,4,5,11], Qi Zheng [1,2,11], Yuqing Xing[1,2,11], Hui Chen [1,2], Yan Li[1,2], Zhixin Hu [6], Shiyu Zhu [1,2], Jingsi Qiao [4,7], Yu-Yang Zhang [2], Haixia Cheng[4], Zhihai Cheng [4], Xianggang Qiu[1,2], Enke Liu [1,2], Hechang Lei [4], Xiao Lin[2], Ziqiang Wang[8], Haitao Yang [1,2] ✉, Wei Ji [4,9] ✉ & Hong-Jun Gao [1,2,10] ✉

Kagome-lattice materials possess attractive properties for quantum computing applications, but their synthesis remains challenging. Herein, based on the compelling identification of the two cleavable surfaces of Co₃Sn₂S₂, we show surface kagome electronic states (SKESs) on a Sn-terminated triangular Co₃Sn₂S₂ surface. Such SKESs are imprinted by vertical *p-d* electronic hybridization between the surface Sn (subsurface S) atoms and the buried Co kagome-lattice network in the Co₃Sn layer under the surface. Owing to the subsequent lateral hybridization of the Sn and S atoms in a corner-sharing manner, the kagome symmetry and topological electronic properties of the Co₃Sn layer is proximate to the Sn surface. The SKESs and both hybridizations were verified via qPlus non-contact atomic force microscopy (nc-AFM) and density functional theory calculations. The construction of SKESs with tunable properties can be achieved by the atomic substitution of surface Sn (subsurface S) with other group III-V elements (Se or Te), which was demonstrated theoretically. This work exhibits the powerful capacity of nc-AFM in characterizing localized topological states and reveals the strategy for synthesis of large-area transition-metal-based kagome-lattice materials using conventional surface deposition techniques.

Transition metal (TM)-based kagome materials provide an attractive platform for investigating correlated topological properties[1–15] and developing kagome-lattice applications[16]. However, the number of already synthesized kagome materials is limited, thus the construction of tailored kagome electronic band structures remains challenging[12]. Kagome physics is generally two-dimensional (2D) but kagome electronic states (KESs) are usually induced by kagome planes embedded in the bulk material. These planes of most known TM-based kagome materials are buried under cleavable surfaces of these crystals[14,17] because the kagome planes usually exhibit higher surface energies than their vicinal planes do[10], which is a roadblock

for direct observation and manipulation of KESs in these materials. This issue, however, can be addressed if we are able to build KESs on a surface of relatively low surface energy (termed surface kagome electronic states, SKESs). SKESs refer to surface electronic states that their wave functions are spatially distributed following the kagome symmetry, namely in a honeycomb lattice where the nodal points of the lattice are comprised of corner-sharing triangles. Building the SKESs enables direct real-space observation and manipulation of rich kagome physics in the atomic limit by surface characterization techniques, such as scanning tunneling microscopy (STM) and atomic force microscopy (AFM). As a consequence, feasible

---

tunability of the SKESs could also be achieved by tip manipulation or molecular beam epitaxy.

Introduction of a capping layer composed of nonmetal atoms over a kagome plane often lowers its surface energy[10]. A SKES is thus built if the capping layer could inherit the underlying kagome symmetry electronically. $Co_3Sn_2S_2$, a kagome magnetic Weyl semimetal, is a good candidate to verify whether SKESs could appear on a non-kagome-lattice surface of a kagome crystal. Two major types (Type-I and Type-II) of its surfaces show triangular patterns in STM images[10,17–21] but kagome-related electronic properties could be observed on vacancies of the Type-I surface and the pristine Type-II surface[17,21]. Meanwhile, the characteristic for Weyl fermions is also different on these surfaces. The Type-II surface shows Weyl Fermi arcs contours, whereas on the Type-I surface, the Fermi arcs is absent[10]. The triangular patterns observed in STM images and a comparison of theoretical surface energies indicate that these two surfaces are, most likely, not $Co_3Sn$-terminated ones, but are covered by S or/and Sn atoms. However, it is still in debate the identification of Type-I and Type-II surfaces and it is yet to be clarified the reason why the triangularly-appeared Type-II surface could exhibit kagome-related properties.

Herein, we experimentally and theoretically verified our inference for building SKESs on $Co_3Sn_2S_2$ surfaces and theoretically tested to tune the SKES by substituting surface Sn or S atoms. SKESs were observed in non-contact atomic force microscopy (nc-AFM) images on the Sn surface. While on S surface, the surface atoms also inherit the triangular symmetry of the $Co_3$ trimer underneath, forming incomplete SKESs (i-SKESs), in which negligible electron hopping exists in one of the two kagome triangles. Force spectra and associated density functional theory (DFT) calculations were used to confirm the assignment of type-II (-I) surface to the Sn- (S-) terminated surface. The nc-AFM images were interpreted with the help of DFT calculations, which reveal that the strong $p–d$ hybridization of the $p$ orbitals of surface Sn or subsurface S with the $d$ orbitals of the $Co_3$ kagome network

underneath occurs near the Fermi level. The hybridization leads to an SKES (i-SKES) imprinted on the Sn-terminated Type-II (S-terminated Type-I) surface. We also theoretically generalized this SKES construction strategy by substituting the surface Sn (S) atoms with group III-A, IV-A, or V-A element (Se or Te) atoms.

## Results

### SKESs on Type-II surface of $Co_3Sn_2S_2$

The bulk crystal of $Co_3Sn_2S_2$ belongs to space group $R\bar{3}m$, comprising a triangular lattice with constants $a = 5.37$ Å and $c = 13.15$ Å[11]. As shown in Fig. 1a and Supplementary Fig. 1, $Co_3Sn_2S_2$ has a layered structure composed of a kagome $Co_3Sn$ plane (red atoms) sandwiched between two triangular S planes (yellow atoms), which are then further encapsulated by two separate triangular Sn planes (blue atoms). Unlike STM images, which provide information on delocalized electronic states, qPlus nc-AFM images could reveal the spatial gradients of short-range repulsive interactions resulting from localized electronic states at the single-chemical-bond level[22–31]. Consequently, it can trace electronic orbitals and/or interactions with unprecedented resolution[22,32]. Therefore, we performed nc-AFM imaging and short-range force spectroscopy on cleaved $Co_3Sn_2S_2$ surfaces. The AFM was equipped with a qPlus sensor, and a CO-functionalized tip was used for imaging (Fig. 1b).

There are two commonly obtained cleaved $Co_3Sn_2S_2$ surfaces: one consisting of a few vacancies (Type-I) and the other containing many adatoms (Type-II). By local contact potential difference (LCPD) measurement, the Type-I surface, with a much higher work function, is demonstrated most likely to be the S surface[21]. However, the $Co_3Sn$ and Sn surfaces have comparable surface work functions and cannot be convincingly distinguished by LCPD, thus leaving the Type-II surface identification a remaining issue. Figure 1c shows an STM image of the S (Type-I) surface, which features a triangular lattice with a few S vacancies. In the nc-AFM image (Fig. 1d), the surface S atoms appeared as bright spots with slightly anisotropy (region $\alpha_I$), maintaining their

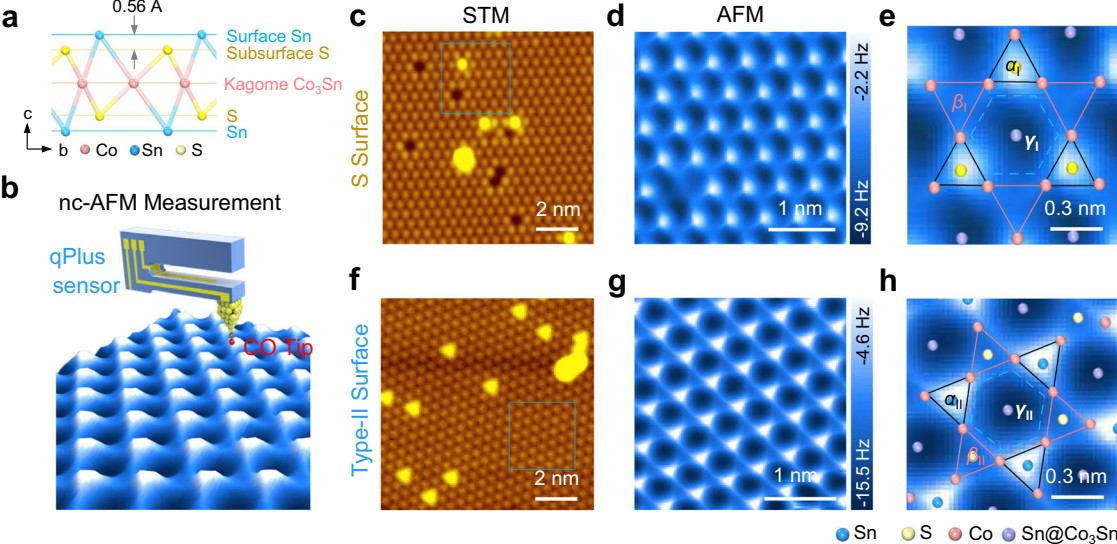

**Fig. 1 | Nc-AFM images of the S surface and Type-II surface in $Co_3Sn_2S_2$. a** Side view of the atomic model of the vertically stacked Sn-S-$Co_3Sn$-S-Sn layers in $Co_3Sn_2S_2$. **b** Schematic of the nc-AFM measurements using a qPlus sensor with a CO-functionalized tip. **c** STM image of the S surface of $Co_3Sn_2S_2$. **d** Chemical-bond-resolved nc-AFM image of the S surface taken in the blue square in (**c**). **e** Zoomed-in image from (**d**) to show the incomplete kagome pattern. Three distinct regions within a unit cell with bright, blurry, and dark contrast, which are marked by black solid line triangles, red solid line triangles, and a blue dashed line hexagon, are labeled as $\alpha_I$, $\beta_I$, and $\gamma_I$ regions. The atomic structure of S surface with the underlying $Co_3Sn$ plane is superimposed. **f** STM image of the Type-II surface of $Co_3Sn_2S_2$.

**g** Chemical-bond-resolved nc-AFM image of the Type-II surface taken in the area marked by a blue square in (**f**). **h** Zoomed-in image from (**g**), showing the kagome pattern. Three distinct regions within a unit cell with bright, blurry, and dark contrast, which are marked by black solid line triangles, red solid line triangles, and a blue dashed line hexagon, are labeled as $\alpha_{II}$, $\beta_{II}$, and $\gamma_{II}$ regions. The atomic structure superimposed is the Sn surface with the underlying S and $Co_3Sn$ plane. Scanning parameters: **c**, **f**, $V_s = -400$ mV, $I_t = 100$ pA; **d**, **e**, amplitude = 100 pm, scanning height = 180 pm lower from a tunneling junction of $V_s = -4$ mV, $I_t = 10$ pA; **g**, **h**, amplitude = 100 pm, scanning height = 220 pm lower from a tunneling junction of $V_s = -400$ mV, $I_t = 100$ pA.

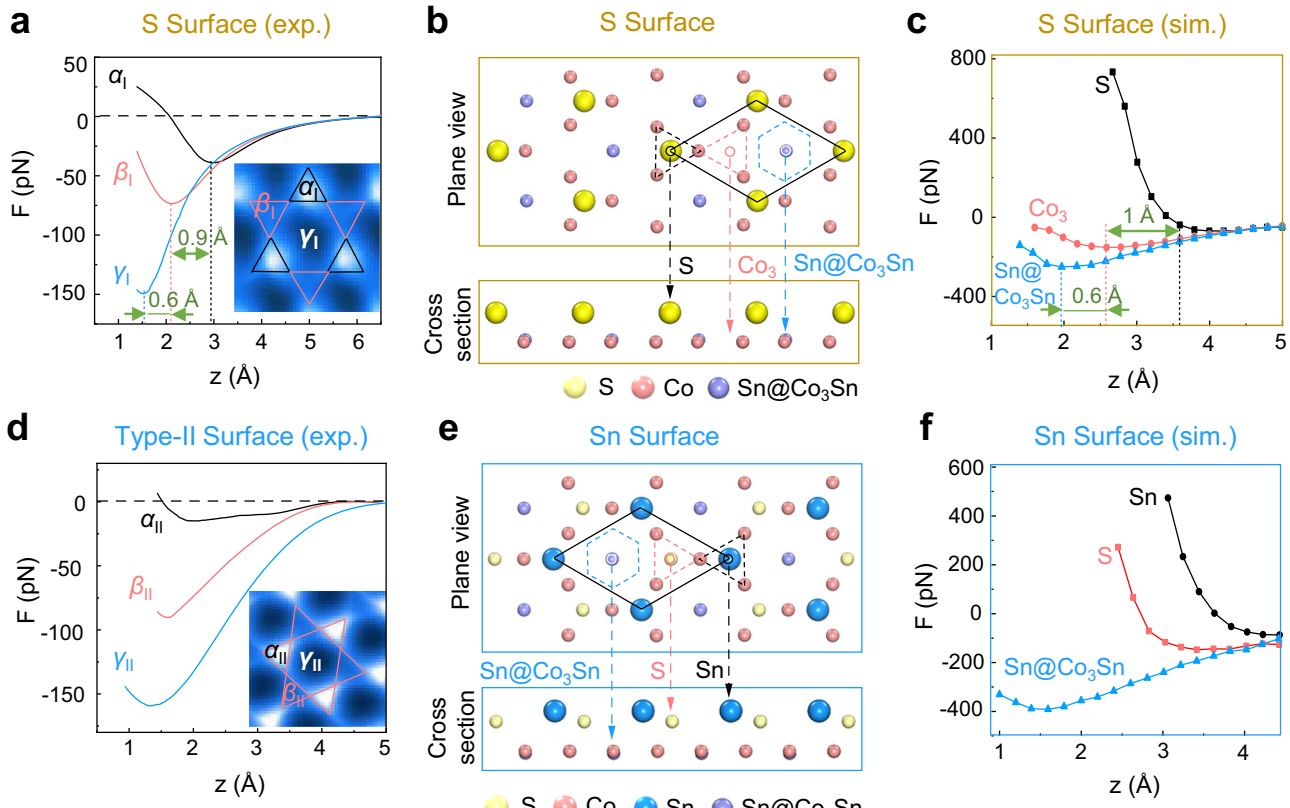

**Fig. 2 | Identification of the Sn surface by short-range force spectra.**
**a** Experimental vertical short-range force spectra measured at the center of the $\alpha_I$ (black), $\beta_I$ (red), and $\gamma_I$ (blue) regions. **b** and **c** DFT optimized surface structure and vertical short-range force spectra on the S surface. **d** Experimental vertical short-range force spectra measured at the center of the $\alpha_{II}$ (black), $\beta_{II}$ (red), and $\gamma_{II}$ (blue) regions. **e** and **f** DFT optimized surface structure and vertical short-range force spectra on the Sn surface. Spectra displayed in (**a**) were obtained with an amplitude of 25 pm and 50 pm for that shown in (**d**). Both experimental vertical short-range force curves were deconvoluted from associated frequency–shift curves using the Sader–Jarvis method[45].

triangular pattern. A further zoomed-in image shown in Fig. 1e shows three line-like features, equivalently distributed in terms of the rotational angle, extending from each S atom. These "lines" converge to rather blurry regions (the $\beta_I$ region) in the image, while the remaining dark region is denoted as $\gamma_I$, implying the formation of i-SKES.

The Type-II surface, which displays a triangular lattice network decorated with a few adatoms in the STM image (Fig. 1f), shows bright dots assembling in the same triangular lattice in the nc-AFM images acquired at a relatively large tip-sample distance (Supplementary Fig. 2a). In a zoomed-in nc-AFM image, we found the dots are circular in shape (region $\alpha_{II}$ marked in Supplementary Fig. 2b), which outline a blurry triangular ($\beta_{II}$), and an indistinctly hexagonal region ($\gamma_{II}$). As the tip approaches the sample surface (Supplementary Fig. 2c, d), the circular shape of the dots evolves into explicitly triangular in the $\alpha_{II}$ region and the blurry triangles become sharper in region $\beta_{II}$. They eventually connect to form a 2D breathing kagome pattern (Fig. 1g). In a zoomed-in image (Fig. 1h), the $\alpha_{II}$ region appears as bright triangles with each side 2.77 Å in length, while the $\beta_{II}$ region appears bigger but as slightly blurry triangles with each side 3.02 Å in length, which is 9% longer than that of the $\alpha_{II}$ region. The kagome-lattice Type-II surface appears to resemble the structure of the $Co_3Sn$ layer, which is, however, less stable compared to the Sn layer under exposure to the surface[10]. We thus meet difficulties in identifying the Type-II surface as either candidates cannot fit both the kagome-appearance and lower-energy criteria. However, if the energetically preferred Sn-terminated surface could display an electronic kagome-appeared pattern, this issue could be resolved by assigning the Sn surface to the Type-II surface. To verify this assumption, we performed force spectra measurements and DFT calculations.

## Identification of Type-II surface

We first measured vertical short-range force spectra on the less controversy S (Type-I) surface to verify the reliability of our experiment and theory comparison. As shown in Fig. 2a, the force spectra at the $\alpha_I$, $\beta_I$, and $\gamma_I$ regions all hit the attractive-repulsive tuning point below 3 Å, with the respective height of the turning point at the $\alpha_I$ ($\gamma_I$) regions 0.9 Å higher (0.6 Å lower) than that at the $\beta_I$ region. In Fig. 2b, we tentatively assessed those three regions to the top-layer S (black dashed triangle), the $Co_3$ trimer (red dashed triangle) and the Sn atom (blue dashed hexagon) in the $Co_3Sn$ layer, respectively. Given that, our DFT calculations well reproduced those experimental force curves in Fig. 2c, which were plotted using the same color code and exhibit comparable respective heights of the turning positions, i.e., 0.9 Å versus 1.0 Å (0.6 Å versus 0.6 Å) for the $\alpha_I$ ($\gamma_I$) region, compellingly supporting those tentative assignments. More importantly, these experiment–theory coincidences, in turn, confirm the assessment of the Type-I surface to the S surface and validate our force spectra comparison method of identifying surface terminations of $Co_3Sn_2S_2$.

Figure 2d plots the experimental force spectra on the Type-II surface, in the same color and spatial schemes used on the Type-I surface. As the tip approaching the surface, the $\alpha_{II}$ region (black) appears the least attractive among all the three regions and first reaches the turning point at 1.92 Å, while that value is 1.60 Å and 1.31 Å for regions $\beta_{II}$, and $\gamma_{II}$, respectively, in which region $\gamma_{II}$ shows a larger attractive force. We modeled the acquiring process of those force curves on both the $Co_3Sn$ and Sn surfaces using DFT calculations. Our primary focus is on the relative positions of the turning points of the three regions, which remain qualitatively unchanged by different tips[29]. For the $Co_3Sn$ surface, the surface Sn atoms show appreciable

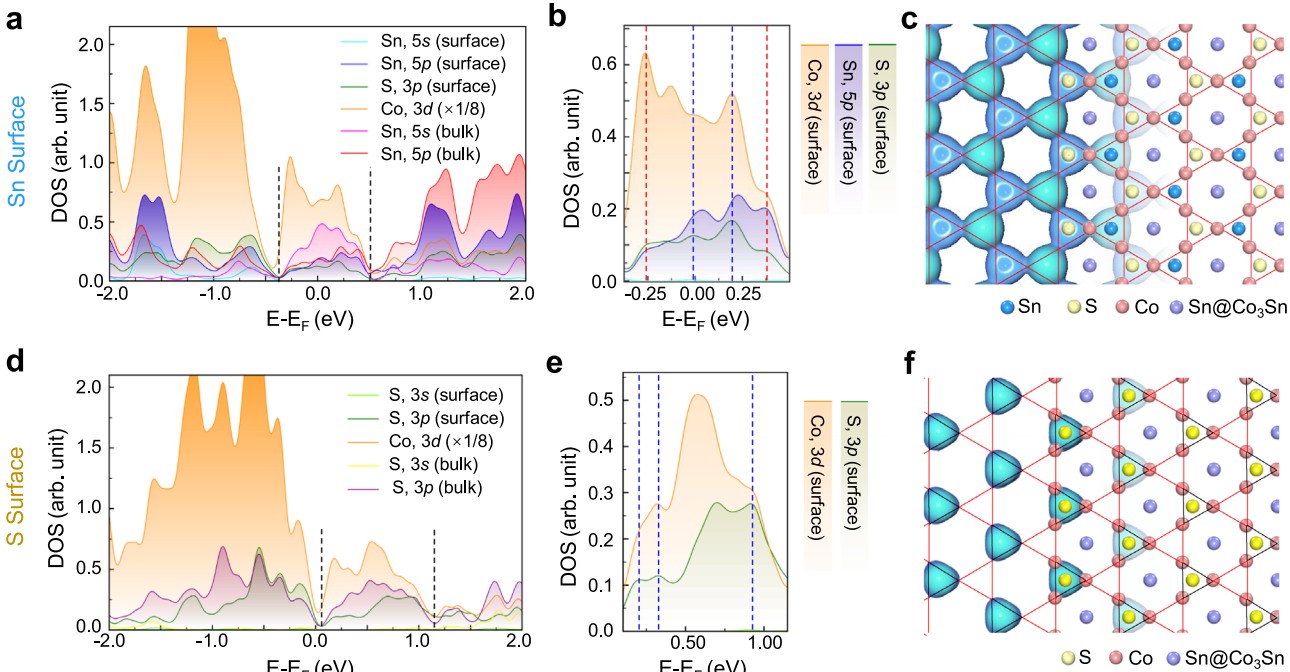

**Fig. 3 | Vertical $p$–$d$ hybridizations on the Sn and S surfaces in $Co_3Sn_2S_2$.**
**a** PLDOS of the Sn surface. Two black dashed lines at −0.38 and 0.50 eV indicate the boundaries of a group of isolated $p$–$d$ hybridized states. **b** PLDOS of the top three atomic layers on the Sn surface, from −0.38 to 0.50 eV. Blue and red dashed lines denote the vertically $p$–$d$ hybridized states between the surface (subsurface) Sn (S) atoms and the $Co_3$ trimers underneath, while the red dashed lines also indicate in-plane (laterally) $p$–$p$ hybridized states between surface Sn and subsurface S atoms. **c** Isosurface contour of $|\psi|^2$, integrated from −0.38 eV to the $E_F$, on the Sn surface

superimposed with the atomic structure of the three topmost atomic layers of the Sn surface in the right part. Red solid lines highlight the kagome pattern.
**d**–**f** Duplicate PLDOS and $|\psi|^2$ isosurface contour plots, in the same schemes used in (**a**–**c**), for the S surface. Boundaries of the isolated $p$–$d$ hybridized states is at 0.12 and 1.16 eV, as marked by the two black dashed lines in (**d**). **e** Zoomed-in PLDOS plotted from 0.12 to 1.16 eV. **f** Isosurface contour of $|\psi|^2$ integrated from 0.12 to 1.16 eV. The black solid line triangles mark the hybridized states on S atoms; while the red solid line triangles mark the missing sublattice of the i-SKES.

surface relaxations on the $Co_3Sn$ surface and are 0.42 Å lifted from the $Co_3Sn$ plane (the cross section in Supplementary Fig. 3a). Consequently, the force spectrum on the Sn atom (blue curve in Supplementary Fig. 3b) first reaches its minimum and shows strong repulsion at shorter distances. The two triangular regions, i.e., the surface $Co_3$ trimer sit on a Sn atom and a S atom underneath ($Co_3$/Sn and $Co_3$/S) are less repulsive in comparison with the surface Sn (hexagonal) region, apparently inconsistent with the experimental spectra on the Type-II surface where the two triangular regions exhibit stronger repulsion. This contradiction does not support the assessment of the $Co_3Sn$ surface to the Type-II surface.

The Sn surface contains triangularly distributed Sn atoms (blue balls in Fig. 2e) in the topmost layer while the same lattice of S atoms (yellow balls), with a lateral shift of (1/3, 1/3) unit cell, is placed in a layer just 0.56 Å below. Each Sn or S atom sits over a $Co_3$ trimer, which could be regarded as a region of triangular symmetry, while the Sn atom in the $Co_3Sn$ layer ($Sn@Co_3Sn$) appears a hexagonal symmetry. Thus, we could tentatively assign the Sn and S triangles to triangular regions $\alpha_{II}$ (black) and $\beta_{II}$ (red), respectively, and the $Sn@Co_3Sn$ hexagon to hexagonal region $\gamma_{II}$ (blue). Associated theoretical force spectra (Fig. 2f) show consistent results with the experimental ones in terms of the order of repulsion among those three regions. This verifies our assignment that region $\alpha_{II}$ ($\beta_{II}$ and $\gamma_{II}$) represents the Sn (S and $Sn@Co_3Sn$) site, as denoted by the superimposed atomic structure in Fig. 1h. Thereby, based on site-dependent force spectra measurements and corresponding DFT calculations, we identified that Type-II surface, which hosts SKESs, is the Sn surface of $Co_3Sn_2S_2$.

The kagome pattern that appeared on the triangular Sn surface is mostly likely to originate from the Co $d$ orbitals, or their hybridized states with surface atoms. Although the local density of states of the Co $d$ orbitals are much larger than that of the Sn/S $p$ orbitals, the $d$ orbitals

are highly localized in real space and electronically screened by the $p$ orbitals of the surface Sn/S atoms. As the repulsive interactions are highly short-ranged in our measurements, the $d$ orbitals in the sub-surface layer should play a rather minor role to the tip-recorded repulsive interactions. Therefore, the detected SKESs should be ascribed to the hybridized states of surface Sn/S atoms rather than directly from the underlying Co $d$ orbitals.

### $p$–$d$ hybridization between surface atoms and the kagome plane

To elucidate the origin of the kagome-shaped feature on the Sn surface revealed in the nc-AFM images in Fig. 1g and h, the projected local density of states (PLDOS) for the Sn-terminated $Co_3Sn_2S_2$ surface is plotted in Fig. 3a. It shows a group of electronic states around the Fermi level ($E_F$), which is nearly isolated from other states and ranges from −0.38 to 0.50 eV, as highlighted by the parallel black dashed lines. These states do not consist of the $s$-orbital component of the surface Sn or S atoms (see Supplementary Fig. 5 for details). We replotted PLDOS between −0.38 and 0.50 eV of the $p$ orbitals of surface Sn and subsurface S atoms and of the $d$ orbitals of the $Co_3$ trimers underneath in Fig. 3b, which indicate strong electronic hybridization between the surface Sn (S) $p$ orbitals, denoted in blue (green), and the $d$ states of the underlying $Co_3$ trimers (in orange). The positions of the four pronounced hybridization peaks are highlighted using four dashed lines.

The $p$–$d$ hybridized states effectively impose the triangular-shaped electron density of the triangular $Co_3$ trimers on those of the surface Sn and S atoms. Figure 3c depicts the square of the wave-function norms ($|\psi|^2$) of the occupied hybridized states, showing a triangular-shaped contour at each Sn or S site. Six of these triangles surround a hexagonal area at the $Sn@Co_3Sn$ site, where the contour density is the lowest, forming a kagome network on the surface. The plot of the $|\psi|^2$ contour also reveals the lateral interconnections of the

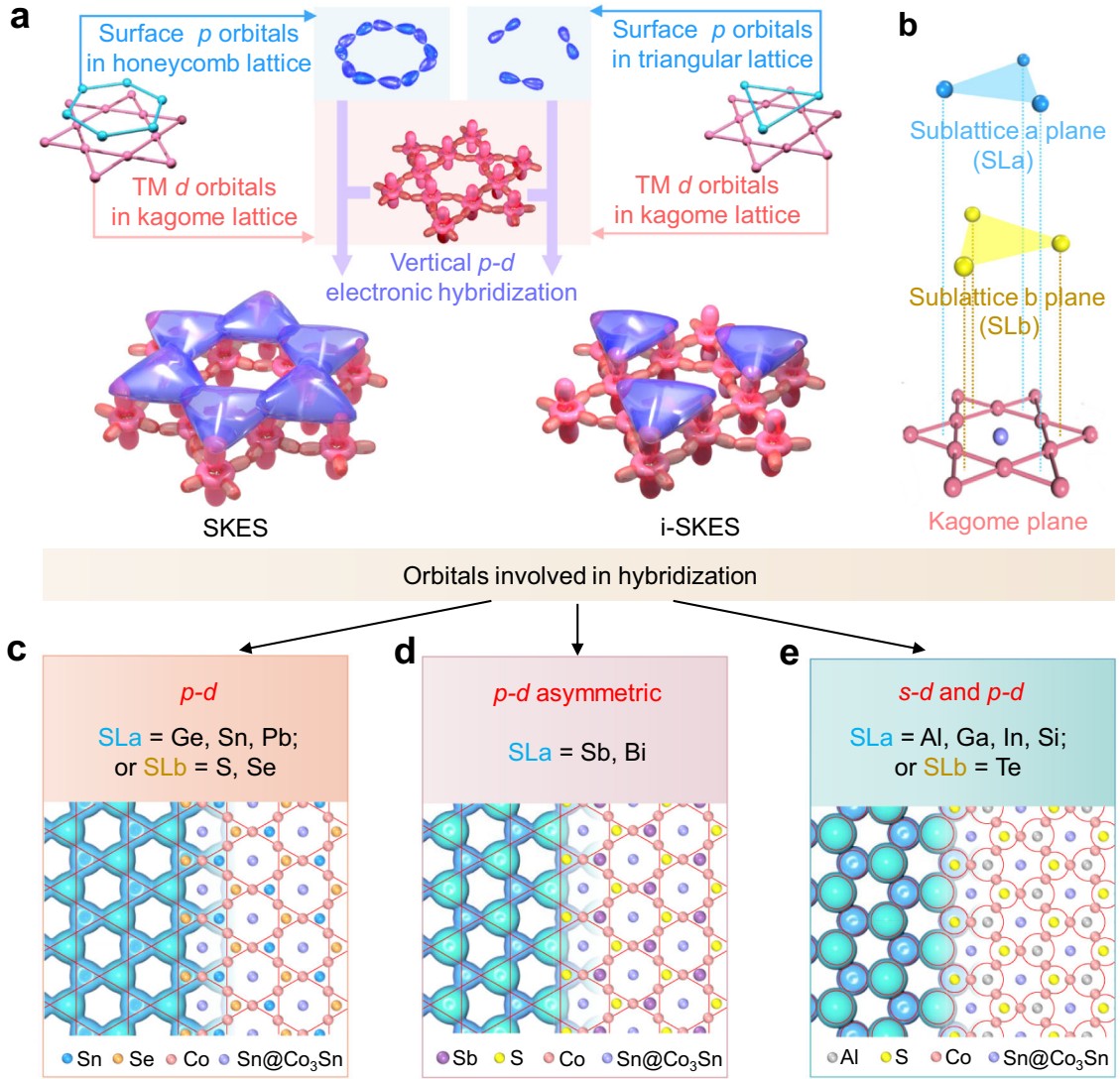

**Fig. 4 | Theoretical strategy for artificially constructing a family of surface kagome electronic states. a** Schematic of SKES (the left part) and i-SKES (the right part) formation through vertical $p$–$d$ hybridization. **b** Schematic of the surface planes of sublattice **a** (blue, SLa plane) and sublattice **b** (yellow, SLb plane) located on top of the kagome plane (red). **c**–**e** Plots of $|\psi|^2$ isosurface contours of hybridized states. **c** Contour of a $p$–$d$ hybridized SKES for SLa = Sn and SLb = Se. Deposition of Ge and Pb on the S surface also generates the same type of SKESs. **d** Contour of a $p$–$d$ hybridized asymmetric SKES for SLa = Sb and SLb = S, where the S and Sb sites show substantially different intensities. Deposition of Bi on the S surface also generates the same type of SKESs. **e** Contour for SLa = Al and SLb = S, where both $s$–$d$ and $p$–$d$ mixed hybridizations are involved. The same type of SKESs can be constructed with deposition of Ga, In and Si on S surface, or substitution of SLb elements with Te.

triangularly shaped density contours, indicating their lateral electronic hybridization in real space. The PLDOS confirm this lateral hybridization between Sn and S, which forms electronic states at −0.25 and 0.37 eV (red dashed lines in Fig. 3b). Therefore, through vertical $p$–$d$ hybridization and subsequent lateral $p$–$p$ hybridization, the kagome symmetry of the Co₃Sn plane underneath is electronically imprinted on the Sn-terminated surface, and an SKES is successfully demonstrated.

The hybridization of Sn and S with the Co kagome-lattice network also transfers the nontrivial properties of the kagome electronic states onto the Sn surface. A scanning tunneling spectral (STS) measurement of the Sn surface (Supplementary Fig. 6a) exhibits a sharp peak centered at approximately −6 meV ($P_k$), as indicated by the red arrow, which fits well with the flat band near $E_F$ in the theoretical band structure of the Sn surface (Supplementary Fig. 6b). The band structures projected on surface $p$ orbitals for Sn surface exhibit one flat band and its crossing with a quadratic band at the Γ point, as illustrated in Supplementary Fig. 6c, demonstrating that the SKESs have the

characteristic feature of the kagome band structures. The position of $P_k$ shifts toward $E_F$ under an applied magnetic field perpendicular to the surface with both up- and down-field orientations (Supplementary Fig. 6d). Such an unconventional splitting trend indicates negative flat band magnetism, further confirming the topological origin of the SKESs.

DFT calculation results for the S surface are displayed in Fig. 3d–f for comparison with those of the Sn surface. The associated PLDOS spectra were plotted in Fig. 3d. It exhibits a group of isolated states between 0.12 and 1.16 eV, where only the $p$ orbitals of S (both at surface and in bulk) and $d$ orbitals of Co are involved. In this energy window, a zoomed-in PLDOS plot of the surface S and subsurface Co₃Sn layers, also indicates vertical hybridization between the surface S $p$ states and the underlayer Co $d$ states, denoted by the dashed lines in Fig. 3e. These vertical $p$–$d$ hybridizations impose triangular-shaped Co₃ electronic states on the surface S atoms, as revealed by the $|\psi|^2$ contour in Fig. 3f, which could be regarded as an i-SKES because there are no electronic states on top of the other sublattice over the Co kagome

network (marked by red solid line triangles in Fig. 3f). As a result, the flat bands observed around $E_F$ on the Sn surface were absent on the S surface, even in a wider energy window[10,21].

## Strategy for constructing a family of SKESs

We have demonstrated the feasibility of SKES construction on the Sn surface of $Co_3Sn_2S_2$ by nc-AFM measurements and DFT calculations. The construction strategy requires that (i) the surface and subsurface (if any) atoms fit in a honeycomb lattice, (ii) their in-plane states vertically hybridize with both of the corner-sharing triangular-shaped sublattices of the kagome lattice underneath, and (iii) their hybridized states then subsequently hybridize laterally with each other to form an SKES, as illustrated in the left part of Fig. 4a. In $Co_3Sn_2S_2$, the Sn-terminated surface approximately meets all these requirements, in which the surface Sn atoms are located on one sublattice, and subsurface S atoms, only 0.56 Å below the surface, reside on the other sublattice.

On the other hand, the S-terminated surface in $Co_3Sn_2S_2$ does not meet requirement (i), thus can only form an i-SKES (see the right part of Fig. 4a). But the additional deposition of hetero- or homo-atoms onto the other sublattice (sublattice plane **a**, SLa, see Fig. 4b) fills the incomplete part of the i-SKES, enabling the construction of SKESs on the S surface with tunable properties. Furthermore, we could obtain more diverse types of SKESs by substituting the S atoms with Se or Te (sublattice plane **b**, SLb, see Fig. 4b) during bulk crystal growth.

DFT calculations were performed to examine the geometric and electronic structures of the surfaces with modified SLa and SLb planes. Diverse surface electronic hybridizations and SKESs are verified. Similar vertical $p-d$ electronic hybridizations are preserved when SLa = Ge, Sn, or Pb and SLb = S or Se. A typical $|\psi|^2$ contour of the isolated states near $E_F$ for SLa = Sn and SLb = Se is shown in Fig. 4c, revealing kagome features that are comparable to those of the Sn-terminated $Co_3Sn_2S_2$ surface. The PLDOS (Supplementary Fig. 7a) indicates that the hybridization states comprising Sn and Se $p$ orbitals and $d$ orbitals of Co are restricted within the energy range of −0.5 to 0.5 eV. They are isolated from other electronic states and can therefore be considered pure $p-d$ hybridized SKES near $E_F$. We also constructed a more typical breathing SKES by significantly unbalancing the hybridized states distributed on the SLa and SLb sites, as achieved by substituting the SLa site with heavier elements, like Sb and Bi (Fig. 4d and Supplementary Fig. 7b). For SLa = Al, Ga, In, or Si or SLb = Te, their $s$ orbitals, in addition to the $p-d$ hybridization, are involved in vertical hybridization with Co around $E_F$ (Fig. 4e and Supplementary Fig. 7c). The involvement of the $s$ orbital results in the $|\psi|^2$ contour of the hybridized states appearing spherical in shape, which significantly weakens the kagome feature on the surface. The energetically isolated hybridized states are mixed with other states for surfaces where SLa = P or As, in which the surface kagome feature is eliminated (Supplementary Fig. 7d). These results reveal that the S surface of $Co_3Sn_2S_2$ can be used as a template for constructing various SKESs with tailored properties.

## Discussion

In summary, we have demonstrated a strategy for constructing energetically stable SKESs that involves electronically imprinting the kagome-lattice symmetry of an undersurface layer through vertical $p-d$ and lateral $p-p$ electronic hybridizations. The strategy was verified using the Sn surface of $Co_3Sn_2S_2$, which is a prototypical kagome magnetic Weyl semimetal. By combining chemical-bond-resolved nc-AFM images, vertical short-range force spectra, and DFT calculations, we explicitly found SKESs on the Sn surface of $Co_3Sn_2S_2$. Our DFT calculations revealed that these states originate from the strong vertical electronic hybridization between triangularly arranged surface Sn or subsurface S atoms and the $Co_3$ trimers of the $Co_3Sn$ kagome lattice underneath, accompanied by lateral hybridizations of the surface Sn

and subsurface S states. STS measurements verified the appearance of kagome symmetry featured topological properties on the Sn surface. To generalize this strategy, DFT calculations explored the surface electronic structure variations for atomic substitutions of surface Sn (subsurface S) with III-A, V-A, and IV-A (Se or Te) elements. We theoretically demonstrated the feasibility of constructing surface kagome structures by depositing group III-A or IV-A elements on the S surface. We also noticed some recent reports on other kagome materials, such as $AV_3Sb_5$ and $RV_6Sn_6$, show the typical kagome bands on honeycomb lattice surfaces[14,33–36], which coincide with the proposed mechanism in this work. The demonstration of nc-AFM for investigating localized electronic surface states aside, our strategy for building SKESs enriches the design routes to construct semi-metallic 2D kagome materials with topologically nontrivial properties.

## Methods

### Single-crystal growth of $Co_3Sn_2S_2$

A Sn/Pb mixed flux was used to grow single crystals of $Co_3Sn_2S_2$. First, the starting materials were mixed at a molar ratio of Co:S:Sn:Pb = 12:8:35:45 (Co (99.95% Alfa), Sn (99.999% Alfa), S (99.999% Alfa), and Pb (99.999% Alfa)). The mixture was placed in an $Al_2O_3$ crucible sealed in a quartz tube that was slowly heated to 673 K for 6 h and then maintained for 6 h to avoid heavy loss of sulfur. Thereafter, the quartz tube was heated to 1323 K for 6 h and maintained for 6 h before slowly cooling to 973 K over 70 h. At 973 K, rapid decanting and subsequent spinning in a centrifuge were performed to remove flux. Finally, hexagonal-plate single crystals with diameters of 2–5 mm were obtained. Energy-dispersive X-ray spectroscopy and X-ray diffraction were used to determine the compositions and phase structures of the crystals.

### QPlus nc-AFM measurements

Non-contact AFM measurements were performed on a combined nc-AFM/STM system (Createc) at LHe temperature with a base pressure lower than $2 \times 10^{-10}$ mbar. All measurements were performed using a commercial qPlus tuning fork sensor[37] in the frequency modulation mode with a Pt/Ir tip at 4.5 K. The resonance frequency of the AFM tuning fork was 27.9 kHz, and the stiffness was approximately 1800 N/m. The $Co_3Sn_2S_2$ samples were cleaved at <10 K in an ultrahigh vacuum chamber and transferred to the nc-AFM/STM head within 10 s. nc-AFM images and short-range force spectra were recorded using CO-functionalized tips. The oscillation amplitudes used in all measurements are stated in the corresponding figure captions.

### DFT calculations

DFT calculations were performed using the generalized gradient approximation for the exchange-correlation potential, the projector augmented wave method[38] and a plane-wave basis set as implemented in the Vienna ab-initio simulation package[39,40]. The energy cutoff for the plane-wave basis was set to 400 eV for structural relaxations and 500 eV for the energy and electronic structure calculations. Two $k$-meshes of $7 \times 7 \times 1$ and $11 \times 11 \times 1$ were adopted for the structural relaxations and total energy (electronic structure) calculations, respectively. The mesh density of the k points was fixed when performing the related calculations with primitive cells. In geometric structure relaxation, van der Waals (vdW) interactions were considered at the vdW-DF level with the optB86b functional as the exchange functional (optB86b-vdW)[41,42]. Symmetrical slab models were employed, and the surface atoms were fully relaxed until the residual force per atom was less than 0.005 eV Å$^{-1}$. To avoid image interactions between adjacent unit cells, a vacuum layer of more than 20 Å thick was added to the slab cell perpendicular to the surface. The optimized lattice constants of bulk $Co_3Sn_2S_2$ are 5.37 and 13.15 Å along the $a$ and $c$ directions, respectively. A tip-sample model was established for the force spectral calculations. In particular, the AFM tip was

modeled using a five-layer thick Pt(111) cluster, the bottom of which adsorbs a CO molecule, and a $(4 \times 4)$ $Co_3Sn_2S_2$ slab was employed to model the sample surface. Our tip model exhibited a significant p-wave feature, which contributes to the high resolution in images acquired using CO-functionalized tips[43,44]. Energetically favorable magnetic ground states were calculated and used for all the tip-sample configurations.

## Data availability

All data that support the findings of this study are present in the paper and the Supplementary Information. Further information can be acquired from the corresponding authors upon reasonable request.

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

## Acknowledgements

The work is supported by grants from the National Natural Science Foundation of China (61888102 (H.-J.G.), 61761166009 (W.J.), 11974422

(W.J.), and 12104313 (X.K.)), the National Key Research and Development Projects of China (2019YFA0308500 (H.-J.G. and L.H.), 2018YFE0202700 (W.J.), 2022YFA1204100 (H.Y. and H.Chen)and 2018YFA0305800 (L.H., Y.Z., and X.L.)), the Chinese Academy of Sciences (XDB30000000 (W.J., L.H. and H.Chen), YSBR-003 (L.H., H.Chen and X.L.)), and the Innovation Program of Quantum Science and Technology (2021ZD0302700 (H.-J.G., H.Y., L.H and H.Chen)). W.J. gratefully acknowledges the Fundamental Research Funds for the Central Universities of China and the Research Funds of Renmin University of China (22XNKJ30). X.K. gratefully acknowledges funding by the Department of Science and Technology of Guangdong Province grant 2021QN02L820 and the Shenzhen Natural Science Fund (the Stable Support Plan Program 20220810161616001). Z.W. is supported by the US DOE, Basic Energy Sciences Grant No. DE-FG02-99ER45747. X.K. thanks Prof. Hong Guo for his financial support to the work done at McGill University.

## Author contributions

H.-J.G. and W.J. conceive of this project; L.H., Q.Z., Y.X., H.Chen, Z.C. and X.L. performed nc-AFM/STM experiments with the guidance of H.-J.G.; X.K., Z.H., Y.Z., H.Cheng and W.J. carried out theoretical calculations and analysis; H.Y., X.Q., E.L. and H.L. prepared the samples; Y.L., S.Z. and J.Q. helped in plotting the figures; L.H., Q.Z., X.K., Z.W., W.J. and H.-J.G. write the manuscript with inputs from all authors.

## Competing interests

The authors declare no competing interests.

## Additional information

¹Beijing National Center for Condensed Matter Physics and Institute of Physics, Chinese Academy of Sciences, 100190 Beijing, China. ²School of Physical Sciences, University of Chinese Academy of Sciences, 100190 Beijing, China. ³College of Physics and Optoelectronic Engineering, Shenzhen University, Shenzhen 518060, China. ⁴Beijing Key Laboratory of Optoelectronic Functional Materials & Micro-Nano Devices, Department of Physics, Renmin University of China, 100872 Beijing, China. ⁵Centre for the Physics of Materials and Department of Physics, McGill University, Montreal, QC H3A 2T8, Canada. ⁶Center for Joint Quantum Studies and Department of Physics, Institute of Science, Tianjin University, 300350 Tianjin, China. ⁷MIIT Key Laboratory for Low-Dimensional Quantum Structure and Devices, School of Integrated Circuits and Electronics, Beijing Institute of Technology, 100081 Beijing, China. ⁸Department of Physics, Boston College, Chestnut Hill, MA, USA. ⁹Key Laboratory of Quantum State Construction and Manipulation (Ministry of Education), Renmin University of China, 100872 Beijing, China. ¹⁰Hefei National Laboratory, 230088 Hefei, Anhui, China. ¹¹These authors contributed equally: Li Huang, Xianghua Kong, Qi Zheng, Yuqing Xing. ✉e-mail: htyang@iphy.ac.cn; wji@ruc.edu.cn; hjgao@iphy.ac.cn

