## [Peer Review File · Nature Communications]

REVIEWER COMMENTS

Reviewer #1 (Remarks to the Author):

In this manuscript, the authors report the discovery of surface kagome electronic states (SKESs) on the Sn surface of $\text{Co}_3\text{Sn}_2\text{S}_2$ combined with non-contact AFM and DFT. Moreover, they tune the SKESs by substitute transition metal atoms on the Sn and S surface in theoretically. This study will help to further understanding the physical properties of kagome lattice and provide a new strategy to construct novel kagome material. The work is undoubtedly interesting and important for kagome materials with novel properties. Thus, I think that this manuscript deserves publication in Nature Communications. However, before the publication, I suggest that the authors should consider the following comments:

1) In STS experiment of Sn surface, does it show a sharp peak at -10 meV or ~ 10 meV?

The information is different between main text and extended data. In addition, from the band structure of Sn surface in extended data Fig. 6b, it only shows one flat band near EF.

2) The authors concluded that the hybridization of Sn and S with Co kagome layer can transferred nontrivial properties of kagome lattice onto Sn surface. Except for the flat band, is there any characteristic for Weyl fermions in experiment or band structure on Sn surface?

Reviewer #2 (Remarks to the Author):

This article focused on the charge-transfer kagome metal $\text{Co}_3\text{Sn}_2\text{S}_2$ and its Type-I (S-terminated) and Type-II surfaces with triangular lattices and kagome-related electronic properties. The authors successfully identified the Type-II surface as the Sn-terminated surface according to nc-AFM imaging, short-range force spectra, and DFT simulations. Then, by the PLDOS and $|\psi|^2$ isosurface contour plots of the Sn surface and S surface, they further demonstrated that the reason for the formation of a surface kagome electronic state by electronically imprinting the below kagome lattice is the vertical p-d hybridization between Co and S/Sn and the subsequent lateral p-p hybridization between S and Sn. At last, the authors summarized their strategy for constructing SKESs and examined this method on various surface elements by DFT calculations. I believe this paper deduced

a logical result with convincing experiments and theoretical calculations, though the result is mundane and not surprising. I support its publication with a few reservations.

First, a generalized conclusion is drawn from a single material. It could be better if authors could experimentally prove their strategy on different materials to make the results more solid. Specifically, experimental evidence of different materials with (or without) p-d hybridization material showing the presence (or absence) of kagome lattice will validate the essence of the p-d hybridization.

Furthermore, The presentation of the current manuscript is in a scattered fashion. The focus appears to be two-folded. Figures 1 and 4 elucidate the importance of p-d hybridization, while the middle figures mainly concern the chemical termination of the Type II surface. It is my personal preference to have Figure 1 move to the last one to show the theoretical explanations after experiments. For now, the connection between Figure 1 and the rest of the manuscript is unclear to me.

Last but not least, I notice the following types in the manuscript:

P3: "These plans of most known TM-based kagome materials are buried under cleavable surfaces of these crystals": plansplanes.

P6: Caption of Fig. 2: (a) and (b) should be reversed.

P8: "(d). Both experimental vertical short-range force curves were deconvoluted from associated frequency-shift curves using the Sader-Jarvis method": frequency-shitfrequency-shift.

P10: "Figure 3c depicts the square of the wavefunction norms ($|\psi|^2$) of the occupied hybridized states, showing a triangular-shaped contour at each Sn or S site": Figure 3cFigure 4c.

Reviewer #3 (Remarks to the Author):

Kagome physics is in the frontier of condensed matter research. However, exposure of the kagome electronic states from the bulk kagome materials is not easy due to the less accessibility of the kagome terminations. By employing the microscopy techniques combined with DFT calculations, the authors demonstrated that the kagome electronic states could be manifested indirectly through the hybridization between d and p orbitals of different sublattices on the more feasible non-kagome terminations. Exemplified by the kagome Weyl semimetal $\text{Co}_3\text{Sn}_2\text{S}_2$, they found that the Sn-terminated surface, which can be regarded as a bucked honey surface, shows electronic states with all symmetries of the kagome lattice, owing to the hybridization between Co-d and Sn/S-p orbitals. In contrast, due to the absence of lateral p-p hybridization, the S-terminated triangular surface shows more triangular electronic states. Thus they proposed to control the surface kagome electronic states by controlling the orbital hybridization between different atoms by surface sublattice substitution. This work brings the kagome electronic states to the surface for easier manipulation, which is crucial for further exploration of rich properties in kagome materials in real

space. Considering the potential significance of this work on kagome physics, I think it deserves publishing in Nature Communications.

Additional comments.

1. According to the authors' proposal, the d-orbital kagome electronic states of the underlying kagome sublattice are actually manifested by the p orbitals of the Sn/S p orbitals. The local density of states of the d orbital is much larger than that of the p orbitals in the hybridization energy range. In the experiment, the AFM tip is much closer to the Sn/S atoms than to the kagome sublayer. Are the detected surface kagome electronic states possibly still mainly from the kagome sublattice? If possible, what's the role of p-d hybridization and why is it needed for the kagome states? If not, how can the authors distinguish between the d and p states for their detected surface kagome electronic states? Does the surface p-orbital-contributed band structure resemble the characteristic kagome band structure?

2. I noticed many recent experiments on kagome materials, e.g., the RV6Sn6 and related materials, showing the typical kagome bands on some surfaces and less prominent kagome bands on others. Does the proposal in this work coincide with the literature? Can the authors exemplify some recent reports to demonstrate the applicability of their proposal to other kagome materials?

3. For the S surface, the forces calculated for the three areas agree well with the experimental ones regarding the turning points of the curves. What could be the reason for the large difference on the Sn surface, especially in the α_{II} and β_{II} regions?

4. The authors claimed on page 7 that "If the kagome-appeared pattern was, as we inferred in Fig. 1c, imprinted electronically from the Co3Sn layer, the Type-II surface would be the energetically preferred Sn-terminated surface." Could the authors explain a little more? I can not follow the logic behind the statement.

5. The concept of "kagome symmetry" imprinted on surface electrons should be defined explicitly.

6. Figure captions for Fig. 2(a) and (b) may be exchanged.

Responses to the comments from the reviewers

Response to Reviewer 1:

In this manuscript, the authors report the discovery of surface kagome electronic states (SKEs) on the Sn surface of $\text{Co}_3\text{Sn}_2\text{S}_2$ combined with non-contact AFM and DFT. Moreover, they tune the SKEs by substitute transition metal atoms on the Sn and S surface in theoretically. This study will help to further understanding the physical properties of kagome lattice and provide a new strategy to construct novel kagome material. The work is undoubtedly interesting and important for kagome materials with novel properties. Thus, I think that this manuscript deserves publication in Nature Communications. However, before the publication, I suggest that the authors should consider the following comments:

We thank the reviewer for the positive remarks on the importance and interest of our work. In the following, we provide our response to the reviewer's comments point-by-point.

Comment 1-1: *In STS experiment of Sn surface, does it show a sharp peak at -10 meV or ~10 meV? The information is different between main text and extended data. In addition, from the band structure of Sn surface in extended data Fig. 6b, it only shows one flat band near E_F .*

Response & Action 1-1: We are sorry for this typo. The peak should be at -6 meV, as shown by the STS in Supplementary Fig. 6a (Extended Data Fig. 6a in the original submission). We revised the figure caption in Supplementary Fig. 6a as follows: "The red arrow marks the flat-band related electronic state residing at -6 mV." Relevant sentences were revised on page 9 of the manuscript as "A scanning tunneling spectral (STS) measurement of the Sn surface (Supplementary Fig. 6a) exhibits a sharp peak centered at approximately -6 meV (P_k), as indicated by the red arrow, which well fits with the flat band feature near E_F in the theoretical band structure of the Sn surface (Supplementary Fig. 6b)."

Comment 1-2: *The authors concluded that the hybridization of Sn and S with Co kagome layer can transferred nontrivial properties of kagome lattice onto Sn surface. Except for the flat band, is there any characteristic for Weyl fermions in experiment or band structure on Sn surface?*

Response 1-2: The characteristic for Weyl fermions is different on Sn and S surfaces. In reference [10] (*Science* **365**, 1286-1291 (2019)), both the Sn- and Co_3Sn - terminated surfaces show Weyl Fermi arcs contours, whereas on the S-terminated surface, the Fermi arcs is absent due to the overlap with the surface-projected bulk bands.

Action 1-2: We added the following description on page 3 in the revised manuscript:

"Meanwhile, the characteristic for Weyl fermions is also different on these surfaces. The Type-II surface shows Weyl Fermi arcs contours, whereas on the Type-I surface, the Fermi arcs is absent¹⁰."

Response to Reviewer 2:

This article focused on the charge-transfer kagome metal $\text{Co}_3\text{Sn}_2\text{S}_2$ and its Type-I (S-terminated) and Type-II surfaces with triangular lattices and kagome-related electronic properties. The authors successfully identified the Type-II surface as the Sn-terminated surface according to nc-AFM imaging, short-range force spectra, and DFT simulations. Then, by the PLDOS and $|\psi|^2$ isosurface contour plots of the Sn surface and S surface, they further demonstrated that the reason for the formation of a surface kagome electronic state by electronically imprinting the below kagome lattice is the vertical p-d hybridization between Co and S/Sn and the subsequent lateral p-p hybridization between S and Sn. At last, the authors summarized their strategy for constructing SKESs and examined this method on various surface elements by DFT calculations. I believe this paper deduced a logical result with convincing experiments and theoretical calculations, though the result is mundane and not surprising. I support its publication with a few reservations.

We thank the reviewer for the thorough review and the positive remarks on the data quality of our work. In the following, we respond to the reviewer's comments point-by-point.

Comment 2-1: *First, a generalized conclusion is drawn from a single material. It could be better if authors could experimentally prove their strategy on different materials to make the results more solid. Specifically, experimental evidence of different materials with (or without) p-d hybridization material showing the presence (or absence) of kagome lattice will validate the essence of the p-d hybridization.*

Response 2-1: We are sorry for the misleading information that the imprint of kagome electronic states by vertical p-d hybridization is a generalized conclusion for many materials. In this work, we proposed this strategy for $\text{Co}_3\text{Sn}_2\text{S}_2$ to interpret the kagome-symmetry-related electronic states on the triangular-lattice Sn plane.

Action 2-1: To rectify the misunderstanding, also to follow the suggestion raised in *Comment 2-2*, we incorporated Fig. 1 into the new Fig.4 and made the following revisions:

We revised the title to restrict the scope of our work: "Discovery and construction of surface kagome electronic states induced by p-d electronic hybridization in $\text{Co}_3\text{Sn}_2\text{S}_2$ ".

On page 2, we revised the following description in the abstract: "Herein, based on the compelling identification of the two cleavable surfaces of $\text{Co}_3\text{Sn}_2\text{S}_2$, we show surface kagome electronic states (SKESs) on a Sn-terminated triangular $\text{Co}_3\text{Sn}_2\text{S}_2$ surface."

On page 11, we revised the first sentence of the section "Strategy for constructing a family of SKESs" as: "We have demonstrated the feasibility of SKES construction on the Sn surface of $\text{Co}_3\text{Sn}_2\text{S}_2$ by nc-AFM measurements and DFT calculations."

Comment 2-2: *Furthermore, the presentation of the current manuscript is in a scattered fashion. The focus appears to be two-folded. Figures 1 and 4 elucidate the importance of p-d hybridization, while the middle figures mainly concern the chemical termination of the Type II surface. It is my personal preference to have Figure 1 move to the last one to show the theoretical explanations after experiments. For now, the connection between Figure 1 and the rest of the manuscript is unclear to*

me.

Response & Action 2-2: We thank the reviewer's constructive suggestion. We agree with the reviewer that the presentation of the manuscript will be improved with Fig. 1 moving to the last one. We merged the original Fig. 1 and Fig. 5 into the new Fig. 4 (Fig. R2) as presented below:

Fig. R2 (new Fig. 4) | Theoretical strategy for artificially constructing a family of surface kagome electronic states. (a) Schematic of SKES (the left part) and i-SKES (the right part) formation through vertical p - d hybridization. (b) Schematic of the surface planes located on top of sublattice a (blue, plane SLa) and sublattice b (yellow, plane SLb) of the kagome plane (red). (c-e) Plots of isosurface contours of wavefunction norms ($|\psi|^2$) for the hybridized states. (c) Contour of a p - d hybridized SKES for SLa = Sn and SLb = Se. Deposition of Ge and Pb on the S surface also forms the same type of SKESs. (d) Contour of a p - d hybridized asymmetric SKES for SLa = Sb and SLb = S, where the S and Sb sites show substantially different intensities. Deposition of Bi on the S surface leads the same type of SKESs. (e) Contour for SLa = Al and SLb = S, where both s - d and p - d and

p-d mixed hybridizations are involved. The same type of SKESs can be constructed with deposition of Ga, In and Si on S surface, or substitution of SLB elements with Te.

Related discussions in the manuscript were revised as follows:

On page 3, the second paragraph of the introduction: “Introduction of a capping layer composed of nonmetal atoms over a kagome plane often lowers its surface energy¹⁰. A SKES is thus built if the capping layer could inherit the underlying kagome symmetry electronically. Co₃Sn₂S₂, a kagome magnetic Weyl semimetal, is a good candidate to verify whether SKESs could appear on a non-kagome-lattice surface of a kagome crystal. Two major types (Type-I and Type-II) of its surfaces show triangular patterns in STM images^{10, 17-21} but kagome-related electronic properties could be observed on vacancies of the Type-I surface and the pristine Type-II surface^{17, 21}. Meanwhile, the characteristic for Weyl fermions is also different on these surfaces. The Type-II surface show Weyl Fermi arcs contours, whereas on the Type-I surface, the Fermi arcs is absent¹⁰. The triangular patterns observed in STM images and a comparison of theoretical surface energies indicate that these two surfaces are, most likely, not Co₃Sn-terminated ones, but are covered by S or/and Sn atoms. However, it is still in debate the identification of Type-I and Type-II surfaces and it is yet to be clarified the reason why the triangularly-appeared Type-II surface could exhibit kagome-related properties.”

On page 4, the third paragraph of the introduction, we added the following description: “SKESs were observed in non-contact atomic force microscopy (nc-AFM) images on the Sn surface. While on S surface, the surface atoms also inherit the triangular symmetry of the Co₃ trimer underneath, forming incomplete SKESs (i-SKESs), in which negligible electron hopping exists in one of the two kagome triangles.”

On page 11, the discussion on Fig. 4a: “(iii) their hybridized states then subsequently hybridize laterally with each other to form an SKES, as illustrated in the left part of Fig. 4a.” and “On the other hand, the S-terminated surface in Co₃Sn₂S₂ does not meet requirement (i), thus can only form an i-SKES (see the right part of Fig. 4a).”

Comment 2-3: Last but not least, I notice the following types in the manuscript:

P3: "These plans of most known TM-based kagome materials are buried under cleavable surfaces of these crystals": plansplanes.

P6: Caption of Fig. 2: (a) and (b) should be reversed.

P8: "(d). Both experimental vertical short-range force curves were deconvoluted from associated frequency-shit curves using the Sader-Jarvis method": frequency-shitfrequency-shift.

P10: "Figure 3c depicts the square of the wavefunction norms ($|\psi|^2$) of the occupied hybridized states, showing a triangular-shaped contour at each Sn or S site": Figure 3cFigure 4c.

Response & Action 2-3: We thank the reviewer for the careful reading and the correction of our manuscript. We corrected the typos and proofread the manuscript throughout.

Response to Reviewer 3:

Kagome physics is in the frontier of condensed matter research. However, exposure of the kagome electronic states from the bulk kagome materials is not easy due to the less accessibility of the kagome terminations. By employing the microscopy techniques combined with DFT calculations, the authors demonstrated that the kagome electronic states could be manifested indirectly through the hybridization between d and p orbitals of different sublattices on the more feasible non-kagome terminations. Exemplified by the kagome Weyl semimetal $\text{Co}_3\text{Sn}_2\text{S}_2$, they found that the Sn-terminated surface, which can be regarded as a bucked honey surface, shows electronic states with all symmetries of the kagome lattice, owing to the hybridization between Co- d and Sn/S- p orbitals. In contrast, due to the absence of lateral p - p hybridization, the S-terminated triangular surface shows more triangular electronic states. Thus they proposed to control the surface kagome electronic states by controlling the orbital hybridization between different atoms by surface sublattice substitution. This work brings the kagome electronic states to the surface for easier manipulation, which is crucial for further exploration of rich properties in kagome materials in real space. Considering the potential significance of this work on kagome physics, I think it deserves publishing in Nature Communications.

We thank the reviewer for the positive remarks on the importance and interest of our work. In the following, we provide our response to the reviewer's comments point-by-point.

Comment 3-1: *According to the authors' proposal, the d -orbital kagome electronic states of the underlying kagome sublattice are actually manifested by the p orbitals of the Sn/S p orbitals. The local density of states of the d orbital is much larger than that of the p orbitals in the hybridization energy range. In the experiment, the AFM tip is much closer to the Sn/S atoms than to the kagome sublayer. Are the detected surface kagome electronic states possibly still mainly from the kagome sublattice? If possible, what's the role of p - d hybridization and why is it needed for the kagome states? If not, how can the authors distinguish between the d and p states for their detected surface kagome electronic states? Does the surface p -orbital-contributed band structure resemble the characteristic kagome band structure?*

Response 3-1: Although the local DOS of the Co d orbitals is much larger than that of the Sn/S p orbitals, the d orbitals are highly localized in real space and electronically screened by the p orbitals of the surface Sn/S atoms. In our measurements, the AFM tip is much closer to the Sn/S atoms than to the kagome Co_3Sn subsurface layer. As the repulsive interactions are highly short-ranged in our measurements, the d orbitals in the subsurface layer should play a rather minor role to the tip-recorded repulsive interactions. Therefore, the detected SKESs were ascribed to the hybridized states of surface Sn/S atoms rather than directly from the underlying Co d orbitals.

In our experiments, we cannot explicitly distinguish the d and p states in the detected SKESs. However, we could find a group of states hybridized by subsurface Co d orbitals and surface Sn/S p orbitals near the Fermi level in our theoretical PDOS, which exhibit the kagome symmetry in real-space distribution that well fits with our nc-AFM images. Therefore, we concluded that the SKESs are imprinted from vertical p - d hybridizations.

The band structures projected on surface- p orbitals (marked with black circles) was plotted in Fig. R3 (Supplementary Fig. 6c) for the Sn-terminated $\text{Co}_3\text{Sn}_2\text{S}_2$ surface. The band structures exhibit a

characteristic feature of the kagome band structures, as depicted with the light-pink straight and parabolic lines: one flat band crosses with another quadratic band at the Γ point. This feature is in excellent accordance with previous reports on the kagome band structures (Nature **584**, 59-63 (2020); Physical Review Letters **99**, 070401 (2007)).

Fig. R3 (Supplementary Fig. 6c) | The band structures projected on surface p orbitals for the Sn surface. A distinct feature of the kagome band structures is resembled, namely the crossing of a flat band and a quadratic band at the Γ point, as depicted with the light-pink straight and the parabolic lines.

Action 3-1: We added the following discussion on page 8 of the revised manuscript:

“The kagome pattern appeared on the triangular Sn surface is mostly likely to originate from the Co d orbitals, or their hybridized states with surface atoms. Although the local density of states of the Co d orbitals are much larger than that of the Sn/S p orbitals, the d orbitals are highly localized in real space and electronically screened by the p orbitals of the surface Sn/S atoms. As the repulsive interactions are highly short-ranged in our measurements, the d orbitals in the subsurface layer should play a rather minor role to the tip-recorded repulsive interactions. Therefore, the detected SKESs should be ascribed to the hybridized states of surface Sn/S atoms rather than directly from the underlying Co d orbitals.”

We added the following discussion on page 9 of the revised manuscript:

“The band structures projected on surface p orbitals for Sn surface exhibit one flat band and its crossing with a quadratic band at the Γ point, as illustrated in Supplementary Fig. 6c, demonstrating that the SKESs have the characteristic feature of the kagome band structures.”

Comment 3-2: *I noticed many recent experiments on kagome materials, e.g., the RV6Sn6 and related materials, showing the typical kagome bands on some surfaces and less prominent kagome bands on others. Does the proposal in this work coincide with the literature? Can the authors exemplify some recent reports to demonstrate the applicability of their proposal to other kagome materials?*

Response & Action 3-2: We thank the reviewer’s constructive suggestion. We added the following discussion and related references on page 13 of the revised manuscript:

“We also noticed some recent reports on other kagome materials, such as AV_3Sb_5 and RV_6Sn_6 , show the typical kagome bands on honeycomb-lattice surfaces^{14, 34-37}, which coincide with the proposed mechanism in this work.”

34. H. Zhao *et al.*, Cascade of correlated electron states in the kagome superconductor CsV_3Sb_5 . *Nature* **599**, 216-221 (2021).
35. S. Peng *et al.*, Realizing Kagome Band Structure in Two-Dimensional Kagome Surface States of RV_6Sn_6 (R=Gd, Ho). *Phys. Rev. Lett.* **127**, 266401 (2021).
36. H. Tan, B. Yan, Abundant lattice instability in kagome metal ScV_6Sn_6 . arXiv:2302.07922 [cond-mat.mtrl-sci] (2023).
37. S. Cheng *et al.*, Nanoscale visualization and spectral fingerprints of the charge order in ScV_6Sn_6 distinct from other kagome metals. arXiv:2302.12227 [cond-mat.str-el] (2023).

Comment 3-3: *For the S surface, the forces calculated for the three areas agree well with the experimental ones regarding the turning points of the curves. What could be the reason for the large difference on the Sn surface, especially in the alpha_II and beta_II regions?*

Response 3-3: As discussed in reference [29] (*Phys. Rev. Lett.* **124**, 096001 (2020)), due to the uncontrollable shape of SPM tip apex, the exact turning points of the short-range force spectra are not the same using different tips, but the order of the turning points along z direction on certain sites is always qualitatively unchanged. On Sn surface, the theoretical force spectra show consistent results with the experimental ones in terms of the order of the turning points of the three regions, which is strong evidence to identify Type-II surface as Sn surface.

Action 3-3: We added the following discussion on page 8 of the revised manuscript:

“Our primary focus is on the relative positions of the turning points of the three regions, which remain qualitatively unchanged by different tips.”

Comment 3-4: *The authors claimed on page 7 that "If the kagome-appeared pattern was, as we inferred in Fig. 1c, imprinted electronically from the Co_3Sn layer, the Type-II surface would be the energetically preferred Sn-terminated surface." Could the authors explain a little more? I cannot follow the logic behind the statement.*

Response & Action 3-4: We are sorry for the unclear expression. We revised it as follows on page 6 of the revised manuscript:

“We thus meet difficulties in identifying the Type-II surface as either candidates cannot fit both the kagome-appearance and lower-energy criteria. However, if the energetically preferred Sn-terminated surface could display an electronic kagome-appeared pattern, this issue could be resolved by assigning the Sn surface to the Type-II surface. To verify this assumption, we performed force spectra measurements and DFT calculations.”

Comment 3-5: *The concept of "kagome symmetry" imprinted on surface electrons should be defined explicitly.*

Response & Action 3-5: We thank the Reviewer for raising this issue. We more clearly explained it in

the first paragraph of introduction on page 3 in the revision:

“SKESs refer to surface electronic states that their wave-functions are spatially distributed following the kagome symmetry, namely in a honeycomb lattice where the nodal points of the lattice are comprised of corner-sharing triangles.”

Comment 3-6: Figure captions for Fig. 2(a) and (b) may be exchanged.

Response & Action 3-6: We thank the reviewer for the careful reading and the correction of our manuscript. We exchanged the figure captions for Fig. 2a and 2b (the revised Fig. 1a and 1b) and proofread the manuscript throughout.

REVIEWERS' COMMENTS

Reviewer #1 (Remarks to the Author):

The authors responded to my concerns and made sufficient clarifications. In my opinion, the paper can now be published.

Reviewer #2 (Remarks to the Author):

The authors have addressed all my previous concerns and I support the publication of this article.

Reviewer #3 (Remarks to the Author):

The revisions throughout make the manuscript much easier to understand for the general audience. All my concerns are removed by the authors and I thus think the manuscript is ready for publication.

Response to Reviewer #1:

Comment: *The authors responded to my concerns and made sufficient clarifications. In my opinion, the paper can now be published.*

Response: We thank the reviewer for the positive remarks and for recommending publication of our work.

Response to Reviewer #2:

Comment: *The authors have addressed all my previous concerns and I support the publication of this article.*

Response: We thank the reviewer for the positive remarks and for recommending publication of our work.

Response to Reviewer #3:

Comment: *The revisions throughout make the manuscript much easier to understand for the general audience. All my concerns are removed by the authors and I thus think the manuscript is ready for publication.*

Response: We thank the reviewer for the positive remarks and for recommending publication of our work.